# Preconditioned MSCs Alleviate Cerebral Ischemia-Reperfusion Injury in Rats by Improving the Neurological Function and the Inhibition of Apoptosis

**DOI:** 10.3390/brainsci12050631

**Published:** 2022-05-11

**Authors:** Jin Zheng, Xueyu Mao, Delong Wang, Shiliang Xia

**Affiliations:** 1Department of Neurology, Tongde Hospital of Zhejiang Province, Hangzhou 310012, China; 2Department of Neurology, Minhang Hospital, Fudan University, Shanghai 201199, China; maoxueyu_mzx@fudan.edu.cn (X.M.); wangdelong900129@fudan.edu.cn (D.W.); 17356432330@sina.cn (S.X.)

**Keywords:** mesenchymal stem cells, neurological function, ischemia and reperfusion, ischemic microenvironment, apoptosis

## Abstract

Mesenchymal stem cells (MSCs) have great application prospects in the treatment of ischemic injury. However, their long-time cultivation before transplantation and poor survival after transplantation greatly limit the therapeutic effect and applications. This study aimed to investigate whether MSCs under the ischemic microenvironment could improve their survival and better alleviate cerebral ischemic injury. Firstly, we used ischemic brain tissue to culture MSCs and evaluated the functional changes of MSCs. Then a middle cerebral artery occlusion (MCAO) model was induced in rats, and the pretreated MSCs were injected via the tail vein. The adhesive removal test, rotarod test, modified neurological severity score, and pathological analyses were applied to assess the rats’ neurological function. Then the expression of neuron and apoptosis related markers was detected. The results indicated that ischemic brain tissue pretreated MSCs promoted the proliferation and the release of the growth factors of MSCs. Meanwhile, in MCAO model rats, transplantation of pretreated MSCs enhanced the neurogenesis, attenuated behavioral changes, reduced infarct size, and inhibited apoptosis. The expression of B-cell lymphoma-2 (Bcl-2), brain-derived neurotrophic factor (BDNF), glial fibrillary acidic protein (GFAP), NF-L, and NeuN were increased, while BCL2-Associated X (Bax) and Caspase-3 decreased. Our results suggest that MSCs pretreatment with stroke brain tissue could be an effective strategy in treating cerebral ischemic injury.

## 1. Introduction

Ischemic stroke is a dominating cause of neuronal dysfunction with high morbidity and mortality [1]. Mesenchymal stem cells (MSCs) have emerged as a potential treatment for ischemic stroke because of their neuroprotective effects, reducing I/R-induced injury, inhibiting I/R-induced apoptosis, and restoring motor function [2,3]. However, their long-time cultivation before transplantation and poor survival after transplantation limit their therapeutic effects and applications. Therefore, developing appropriate methods for improving the efficacy of engrafted MSCs is indispensable.

Bang et al. [4] reported the treatment of patients with severe cerebral infarction (also known as ischemic stroke) by intravenous injection of autologous MSC after 5–6 passages, which for the first time proved the possibility and safety of human MSC in the treatment of human cerebrovascular diseases. Our research team found that MSCs by subculturing twice in vitro had a better therapeutic effect on cerebral ischemia rat models than MSCs that were passaged 6 times. This provides the possibility for clinical autologous MSCs to treat acute ischemic stroke. However, the disadvantage is that the number of MSCs that have been obtained for two passages is limited, and the survival rate of MSCs injected into the rat tail vein in the environment of cerebral ischemia and inflammation is only 8–9%. Pretreatment could enhance the therapeutic ability of MCS [5]. Hypoxia-pretreated MSCs have superior migration ability and viability, which induces increased expression of chemokine receptor type 4 (CXCR4)/7 [6]. Previous studies found that MCS extracted in the later stage of stroke has a better curative effect on stroke rats than in the early stage [7]. In addition, Li reported that different nutritional support has an impact on the repairability of MCS [8] and the nutritional factors in the brain of ischemic stroke rats can promote the treatment of MSC. Additionally, a study found that the exosomes from I/R-injured rat endothelial cells, could significantly improve the ischemic stroke injury of SH-SY5Y cells, down-regulated the expression levels of (Bcl2-Associated X) Bax and Caspase-3, and up-regulate the expression of B-cell lymphoma-2 (Bcl-2) [9]. Based on the above reports and preliminary test results, the goal of this study is to improve the survival rate of MSCs and enhance the paracrine function of MSCs, which is the most critical factor in solving the problem of MSCs in the treatment of acute ischemic stroke. 

Our research team used rat cerebral infarction tissue to pretreat rat MSCs, evaluating the reproductive capacity, survival rate, the levels of nutritional factors, and CXCR4 of the MSCs in vitro. Afterward, the behavioral tests of ischemic stroke rats and measurements of infarct area by TTC were used to evaluate the neuronal injuries of rats and the protective effect of pretreated MSCs. Additionally, the degree of pathological injury and apoptosis in the cerebral infarction area of stroke rats was evaluated by Histopathological Evaluation (HE) and TUNEL staining. The levels of proliferation, differentiation, and injury in neurons were observed by immunohistochemistry. In addition, we detected the apoptosis-related proteins and the nutritional factors by Western blot (WB) to explore the potential mechanism of MSCs to improve ischemic stroke. This paper hopes to afford a valued theoretical basis and experimental reference for the treatment of autologous MSCs transplantation in the acute phase of ischemic stroke.

## 2. Materials and Methods

### 2.1. Animals and Ethics Statement

Sixty-five Sprague-Dawley rats (250 ± 30 g) were acquired from Zhejiang Vital River Laboratory Animal Technology Co., Ltd. (Certificate No. SCXK (Zhe)2019-0001; Jiaxing, China), raised in the Animal Experiment Research Center of Hangzhou Eyong Biotechnological Co., Ltd., Zhejiang, China (Certificate No. SYXK (Zhe)2020-0024). SD rats were raised in standard cages in an environment that had a constant temperature (24 ± 1 °C), constant humidity of 45–55%, and a dark/light cycle of 12 h; in addition, all rats ate and drink freely. Adaptive feeding for one week before the experiment. 

### 2.2. Rat Middle Cerebral Artery Occlusion (MCAO)

The intraluminal suture occlusion technique was used to induce MCAO in the rats [9]. In the ischemia-reperfusion group, the rats were anesthetized with isoflurane (5 percent for induction and 2 percent for maintenance), supine fixation, routine skin preparation, and disinfection. At the median neck incision, the muscle group was separated into the deep part to reveal the right common carotid artery (CCA) and external carotid artery (ECA), with careful separation to avoid damaging the vagus nerve and internal carotid artery (ICA), the pterygopalatine artery was not separated. Additionally, the proximal ends of CCA and ECA were ligated separately. The arterial clip temporarily clamped the distal end of the CCA, the distal end of the CCA was cut into a small opening by ophthalmic scissors, and the prepared nylon thread plug with a diameter of 0.24 mm was inserted along the incision. The hanging thread, placed on the CCA to prevent bleeding, was properly tied. After removing the arteriole clip, the ophthalmic forceps line plug was slowly advanced. When the front of the line-of-sight plug entered the ICA through the CCA bifurcation, the curve of the plug line was adjusted to the lower right direction until there was resistance. The insertion depth was about 18 mm from the CCA bifurcation. The blood vessel would be occluded through the starting end of the MCA and the previous lines were tied to fix the bolt wire. Then the muscle and skin were sutured and the exposed part of the end of the tether was marked. After 3 h of ischemia, the cord was pulled out about 10 mm to make the ball end back into the CCA, which formed reperfusion. The body temperature of the rats was maintained at 37 ± 0.5 °C during ischemia and after reperfusion, and the airway was kept patency by suctioning the secretions at intervals. In the sham group, the application of the anesthesia was the same as that of the ischemia-reperfusion group. Then, the rats had a supine fixation with routine skin preparation and disinfection, the incision was at the median neck and was separated from the muscle group to the deep part to expose the right CCA, ECA, without other treatment, and then the muscle and skin were sutured.

### 2.3. Design of Experimental and Animal Groups

The rats were divided into five groups with six rats at random: (1) Sham: Sham + PBS, (2) MCAO: MCAO + PBS, (3) MCAO + FBS-MSCs: MCAO + FBS-MSCs (2 × 10^6^ cells), (4) MCAO + N-MSCs: MCAO + N-MSCs (2 × 10^6^ cells), (5) MCAO + S-MSCs: MCAO + S-MSCs (2 × 10^6^ cells). One day after MCAO modeling, pretreated MSCs were intravenously injected into the tail veins of the corresponding rats in each group. The sham groups were injected with the same volume of solvent for control.

### 2.4. Behavioral Tests

The behavioral changes of rats with ischemic stroke were observed 1, 7, 14, and 28 days after the tail vein injection of preconditioned MSCs. The behavioral changes were measured by the rotarod test, the adhesive removal test, and Modified Neurological Severity Score (mNSS). 

#### 2.4.1. mNSS

mNSS consists of motor, sensory, reflex, and balance tests that were performed by an investigator blinded to the experimental groups at 1, 7, 14, and 28 d after MCAO. On a scale of 0 to 18, one point means the rats were unable to perform the task or lacked a proper response to a given stimulation [10] Additionally, 9 to 10 are divided into very serious neurological impairment; 7 to 8 are divided into severe neurological impairment; 5 to 6 are divided into moderate neurological impairment; less than 5 is divided into mild neurological impairment.

#### 2.4.2. The Adhesive Removal Tests 

The adhesive removal test, also known as the sensory asymmetry test, is used to examine the skin sensitivity and the integration of sensory functions in mice [11]. Specifically, it is to stick a 6 mm square adhesive label on the ankle of a relatively hairless mouse. In the ordinary mouse, the label of each square was quickly removed with its teeth. While, in rats with cerebral ischemia, the markers that were attached to the normal hind limbs were firstly removed, and the rats spent more time removing the markers on the lesion side because ischemic lesions will lead to sensory asymmetry. In addition, the time delay also indicates a lack of motor function, and dyskinesia of the mouth and hind limbs makes it quite hard to remove the marker.

#### 2.4.3. The Rotarod Test

First, the rats were trained to walk on a rotating tripod with a diameter, width, and height of 5 × 8 × 20 cm at a constant speed of 16 rpm for 1 min. During the training process, each time the rats fell, they would be put back on the fishing rod until the training was completed. After training for 24 h, the rats were tested on a rotating tripod at a constant speed of 20 rpm. Each test consisted of five tests, and each lasted 60 s. The time and the number of times the rats fell from the wheel frame were recorded. Rats that dropped more than five times were excluded from the experiment and were put back into the cage [12].

### 2.5. 2,3,5-Triphenyltetrazolium Chloride (TTC) Staining

The rats were sacrificed 28 days after cerebral infarction. The whole brain was frozen at −20 °C for at least 20 min, then the brain was cut from the frontal lobe about 2 mm thick. Six serial sections of each brain were soaked in 2.0% TTC and incubated in a dark incubator for 30 min at 37 °C. Normal brain tissue-stained is red, while unstained areas are considered damaged areas. The hemispheric infarct volume was measured by Image-J. 

### 2.6. Histopathological Evaluation (HE) Staining

Histopathology of the brain was evaluated by HE. After dewaxing, the sections made of tissue paraffin blocks were stained with HE kits according to the product guidelines and then observed with the microscope (Nikon Eclipse C1, Japan) after sealing. The scoring criteria of lung histopathological injury include four aspects: the degree of alveolar hemorrhage; the degree of hyperemia; the levels of infiltration and the accumulation of neutrophils in the alveolar space and vascular wall; the degree of pulmonary alveolar wall increment in thickness and hyaline membrane formation. It was given marks of 0–4 from light to heavy.

### 2.7. Formation of Neurons in the Infarct Area

Cerebral infarction rats were injected regularly with BrdU (50 mg/kg) into the abdominal cavity for 3 days, and 28 days after cerebral infarction, the brains were taken for paraffin-embedded section, and the BrdU antibody was used for immunohistochemistry by the Computer-Assisted Stereological Toolbox (CAST) system, which counted BrdU positive cells.

### 2.8. Immunohistochemistry 

The sections with paraffin embedding were taken on the 28th day after cerebral infarction, and immunohistochemical staining was used to detect BrdU, neuronal nuclei (NeuN), (glial fibrillary acidic protein) GFAP, and neurofilament light chain (NF-L). Firstly, the rats were anesthetized deeply, the heart was perfused with normal saline and 10% neutral buffered formalin (NBF, pH = 7). After the tissue was processed and paraffin-embedded, a coronal brain slice was made with a thickness of 5 μm slicer (Diapath, Martinengo, Italy) using the following method. Firstly, the slice was deparaffinized and rehydrated and was immersed in a 0.1 M citrate buffer (pH = 6), then the slice was heated for 20 min to recover the antigen. After the super blocking solution (the immunohistochemistry kit, Scytek, USA) was used to block the non-specific antibody binding for 7 min at room temperature, the slice was incubated with BraU (1:1000), NeuN (1:1000), GFAP (1:500), and NF-L (1:1000) antibodies (Abcam, Germany) at 4 °C overnight. For preventing reaction with endogenous peroxidase, the slice was washed and incubated (10 min) in 10% H_2_O_2_. Following this, the slice was treated with a biotin-conjugated secondary antibody. Then the slice was treated with streptavidin peroxidase for 10 min and incubated with diaminobenzidine (DAB) to the color reaction. In the end, the slice was observed with a microscope (Nikon Eclipse Ti-U, Japan). The way of preparing the sections for negative control was the same as above, just without the primary antibody.

### 2.9. TUNEL Staining

After being treated with dimethyl-benzene and absolute alcohol gradients, the section of the brain was covered with protease K (10%). After removing the liquid, the sections were incubated with the TUNEL kit (Servicebio, China), respectively. Finally, it was incubated with DAPI and sealed with an anti-fade mounting medium. DAPI (blue) and positive cells (green) were observed under the fluorescence microscope. Apoptosis rate %: the ratio of TUNEL positive cells to total cells.

### 2.10. MSC Culture

Bone marrow was extracted from rat femurs, and monocytes were separated by density gradient centrifugation with lymphocyte separation solution. The isolated monocytes were cultured with 10 percent fetal bovine serum (FBS) and Dulbecco’s modified Eagle medium (DMEM) containing 1% penicillin/streptomycin. Additionally, the monocytes were passaged twice to obtain MSCs. 

### 2.11. Preconditioning MSCs with Ischemic Brain Tissue and Grouping

The rat ischemic stroke was induced by MCAO surgery as described in Section 2.2. On the 4th day of ischemic stroke, after euthanasia, the rat’s neck fur was cut horizontally, and tissue scissors were used to cut at the junction of the skull and cervical vertebra. Then, the tissue scissors were used to cut from the foramen magnum towards the ipsilateral orbit. The skull was cut between the rat’s eyes and removed to obtain brain tissue. After measuring the brain weight, DMEM was mixed with the brain to 150 mg/mL, and the extract was obtained by squeezing the mixture. Then the supernatant was obtained by centrifuging for 10 min and was mixed with the cell culture medium to cultivate MSC. Then MSCs were divided into three groups: FBS-MSC: MSC + 10% FBS; N-MSC: MSC + 10% normal rat brain; S-MSC: MSC + 10% ischemic rat brain. For the latter two groups, the 10% FBS in the medium was changed to 10% normal brain tissue (N) or 10% ischemic brain tissue (S) to culture cells according to the following grouping. 

### 2.12. MTT Analysis

The MTT assay was used to evaluate cell survival. The cells in logarithmic growth were diluted to 1000 cells/100 μL to inoculate on 96-well plates and were cultured according to the requirements of grouping. Subsequently, the MTT (Beyotime, China) was used following the instruction manual from the manufacturer [13]. After culturing for 4 h, the cells without supernatant were treated with dimethyl sulfoxide (DMSO) for an hour. Lastly, the optical density (OD) of the cell culture medium was read at 490 nm by a microplate reader.

### 2.13. Flow Cytometry Detection

Flow cytometry was used to detect CD105 and CD73, which were the surface markers of MSCs, to identify MSCs. The collected MSCs were washed with PBS, then centrifuged at 1000 rpm for 5 min. The 1 × 10^5^ cells were resuspended in 1 × AnnexinV binding buffer and accompanied by 5 μL of propidium iodide (PI) reagent. After incubating for 15 min in the dark at room temperature, 400 μL of binding buffer was added and the cells were measured by a flow cytometer (Accuri C6, BD Biosciences, NJ, USA).

### 2.14. Enzyme-Linked Immunosorbent Assay

The supernatant of blood-brain tissue cultured MSCs was collected on days 1, 3, 5, and 7, and brain-derived neurotrophic factor (BDNF), (nerve growth factor) NGF, (basic fibroblast growth factor) bFGF, (Hepatocyte Growth Factor) HGF, and vascular endothelial growth factor (VEGF) (R&D System, Minneapolis, MN, USA) were quantitatively immunochemical assayed using a commercially available ELISA kit, following the manufacturer’s instructions. All experiments were repeated independently three times.

### 2.15. Immunofluorescence Detection of Cells

The cells that were cultured on glass slides were fixed with formaldehyde and were permeabilized with 0.5% Triton X-100, then they were incubated with CXCR4 antibody and goat anti-rabbit IgG antibody (Abcam, Germany), respectively. Finally, the cells were counterstained with DAPI and observed under a fluorescent inverted microscope. The positive reaction can be seen as green, and the nucleus was blue.

### 2.16. Western Blotting

The ischemic tissue from the right brain and cells were fragmented for extracting total protein, and the concentration was measured by the BCA kit (Solarbio, China). The protein sample of each group was denatured and diluted to separate on an SDS-PAGE gel by electrophoresis. After the protein sample was transferred to the PVDF membrane by wet transfer method, the membrane was incubated with the diluted primary antibodies (Abcam, Germany) including β-actin (1:1000), BDNF (1:000), GFAP (1:000), Bax (1:000), Bcl-2 (1:000), Caspase3 (1:000), NGF (1:000), bFGF (1:1000), HGF (1:000), and VEGF (1:000) overnight at 4 °C (while shaking). After the PVDF membrane was washed with TBST solution (three times, 5 min each time), the secondary antibody, horseradish peroxidase-conjugated anti-rabbit/mouse immunoglobulin IgG (1:10,000, Abcam, Germany) was used to incubate the membrane for 1 h at normal atmospheric temperature. Lastly, the immune response zone was visualized with enhanced chemiluminescence (ECL) reagents and imaged by the Tanon imaging system (Tanon-4600, Tanon, Shanghai, China). Based on the optical density analysis by ImageJ software, the results were presented with the relative protein expression level, which was the proportion of the gray value of each sample to the gray value of the β-actin.

### 2.17. Statistical Analysis

Statistical analysis that compared the means of two groups was carried out by Student’s t-test and multiple groups were compared using a one-way ANOVA analysis followed by SNK test using SPSS 16.0 software. All data of the paper were shown as mean ± SD. The *p* < 0.05 was considered statistically significant.

## 3. Results

### 3.1. Characteristics of MSCs

In preliminary experiments, after 24 h of primary culture, MSCs appeared as single adherent cells, which were spindle-shaped or polygonal. The cells proliferated rapidly and formed clones within 3–5 days. As shown in Figure 1A, most of the clones were spindle-shaped and multi-protruding fibroblast-like cells. Meanwhile, to determine the difference between the three treatment groups, cell viability was measured by observing the decrease in MTT. It can be seen from Figure 1B that compared with the FBS-MSC group, the cell viability increased significantly with the development of time. Compared with the FBS-MSC group, the S-MSC group and the N-MSC group had higher cell viability at the beginning and end (*p* < 0.001, *p* < 0.01). In addition, the cell viability of the S-MSC group was significantly higher than that of the other two groups (*p* < 0.001). Additionally, flow cytometry was used to detect the levels of MSCs surface markers CD105 and CD73 (Figure 1C). The results showed that all the MSC groups had higher levels of stem cell marker CD105 and CD73 expression.

### 3.2. Comparison of Growth Factor Levels Secreted by MSCs 

We checked the levels of BDNF, NGF, bFGF, HGF, and VEGF in the different treatment groups (ischemic brain tissue treatment group, FBS treatment group, and normal brain tissue treatment group) of MSCs on 1, 3, 5, and 7 days. The nutritional factor level of the s-MSCs group was markedly higher than that in the FBS-MSCs group (*p* < 0.05). The levels of BDNF and HGF proteins expression in the S-MSC group were greater than those in the N-MSC group (*p* < 0.05). The NGF level between N-MSC and S-MSC groups was similar on day 7, while the level of VEGF and bFGF in the N-MSCs group was larger than that in the S-MSC group (Figure 2).

A similar trend was observed in Figure 3A–F. Compared with the FBS-MSCs group, the protein levels of BDNF, NGF, bFGF, and VEGF in the N-MSCs group did not change significantly (*p* > 0.05), the protein levels of HGF was markedly increased (*p* < 0.01). The levels of BDNF, bFGF, HGF, and VEGF proteins in the S-MSCs group were significantly growing (*p* < 0.05 or *p* < 0.01). Compared to the N-MSCs group, the expression levels of BDNF, bFGF, and HGF proteins of the S-MSCs group were significantly growing (*p* < 0.05 or *p* < 0.01).

### 3.3. Comparison of Chemokines of Ischemic Brain Tissue Pretreatment MSCs 

CXCR4, a nonclassical receptor of CXCL12, is expressed on myeloid-derived stem cells, including MSCs derived from bone marrow. The expression intensity of chemokine (CXCR4) in the S-MSCs group was significantly stronger than that in the N-MSCs group and FBS-MSCs group (Figure 4).

### 3.4. S-MSCs Reduced Cerebral Infarction Volume and Attenuates Neurologic Injury in I/R Rats

As shown in Figure 5A,B, infarction was not observed in the sham group by TTC staining, while white infarct lesions appeared in the MCAO and MSCs groups, indicating that the MCAO model was successfully built-in in rats. However, the volume of cerebral infarction in MSCs treatment group significantly decreased (*p* < 0.01). Among them, the reduction in the cerebral infraction area was most significant in the MCAO + S-MSCs group. Next, the behavioral function was evaluated by mNSS, the rotarod test, and the adhesive removal test. Compared with the MCAO group, the mNSS of the MSCs treatment groups at 7 days, 14 days, and 28 days were significantly lower (Figure 5C). Likewise, the tendency to remove adhesive was the same (Figure 5E). Furthermore, compared with the FBS-MSC and N-MSC + MCAO groups, the S-MSC + MCAO group also had a significant improvement in the results of the rotarod test at 7, 14, and 28 days (Figure 5D). In short, our results confirmed that S-MSC injection can improve neurological deficits in I/R rats.

### 3.5. S-MSCs Has Neuroprotective Effects in I/R Rats 

The pathological changes of neurons after CIRI were evaluated by HE stains. The morphology of neurons in the cerebral cortex in the sham group was normal, with clear outlines and well-distributed cytoplasmic staining (Figure 6A,B). The nucleus that was situated in the center of the cell in the brain tissue was visible. The cortical neurons of ischemia-reperfusion were damaged to varying ranges. The neuron was atrophied, blurred, and the cytoplasm was deeply stained. After that, the influence of S-MSCs preconditioning on neuronal apoptosis was observed using TUNEL staining (Figure 6C,D). Comparing the sham group with the MCAO groups, the apoptosis index (AI) in MCAO groups was markedly raised (all *p* < 0.001). However, the AI in both the N-MSCs and S-MSCs groups was markedly reduced than that in the FBS-MSCs group (*p* < 0.001, respectively). Furthermore, the AI of the S-MSCs group was markedly reduced than that of the N-MSCs group (*p* < 0.05). To further demonstrate the neural regeneration phenomenon, BrdU was applied, the BrdU positive cells were blue fluorescently labeled cells, and the positive cells mainly appeared around the infarct (Figure 6E,F). There were only a few positive cells in the sham group, while a large number of positive cells appeared in the MCAO group and MSCs group. Compared with the MCAO group, the degree of BrdU expression in the MSCs group was stronger, the most obvious in the S-MSCs group (*p* < 0.01).

### 3.6. S-MSCs Relief Neuronal and Axonal Injury after MCAO

Compared with the sham group, the protein levels of NF-L and GFAP were greater in the MCAO group (*p* < 0.05 or *p* < 0.01), while the protein level of NeuN was decreased in the MCAO group (*p* < 0.05). Importantly, after the interventional treatment of MSCs, the expression of NeuN, GFAP, and NF-L were markedly changed in the brain tissue of the MSCs treatment (*p* < 0.05 or *p* < 0.01), the most significant change was in the MCAO + S-MSCs group (Figure 7).

Furthermore, we detected the protein levels of BDNF and GFAP with Western Blot, and the results were consistent with those in IHC (Figure 8A,E,F). That is, MSCs reversed the expression levels of BDNF and GFAP, and the treatment effect of the MCAO + S-MSCs group was the most obvious (*p* < 0.01).

### 3.7. S-MSCs Preconditioning Attenuates Cerebral Apoptosis after MCAO

To further explore the action mechanisms of S-MSCs, the relative protein expression levels of apoptosis were examined. The expression degrees of proteins that acted on apoptosis by Western blot, included Caspase-3, Bcl-2, and Bax. The analysis results were shown in Figure 8, compared with those in the sham group, the protein levels of Caspase-3 and Bax were significantly greater in the MCAO group, while the protein level of Bcl-2 was clearly reduced (all *p* < 0.05). The levels of Caspase-3 and Bax proteins in the MCAO + FBS-MSCs, MCAO + N-MSCs and MCAO + S-MSCs group compared to the MCAO group were significantly lesser (*p* < 0.05 or *p* < 0.01). In the MCAO + N-MSCs group, compared to the MCAO + FBS-MSCs group, the levels of Caspase-3 and Bax proteins were reduced (*p* < 0.05). And in the MCAO + S-MSCs group, the degrees of Caspase-3 and Bax proteins were clearly decreased (*p*<0.01). The Bcl-2 protein level was all increased compared to the MSCs treatment group than that in the MCAO group (*p* < 0.01); and it was significantly raised in the MCAO + N-MSCs group compared to the MCAO + FBS-MSCs group (*p* < 0.05); compared with the MCAO + N-MSCs group, the level of it was also greater in the MCAO + S-MSCs group (*p* < 0.05).

## 4. Discussion

In clinical practice, ischemic stroke is still a very tough challenge. Therefore, more and more research focuses on finding effective methods to alleviate cerebral ischemia. Modern research has shown that MSCs therapy positively improved cerebral ischemia/reperfusion damage [14]. The mechanism behind the therapeutic effects of MSC transplantation in the treatment of central nervous system diseases (including ischemia) involves neuroprotection, angiogenesis, neurogenesis, synaptogenesis, and immune regulation of secreting a certain number of neurotrophic factors through MSCs [15,16,17]. However, the effectiveness of MSCs is affected by an unfavorable microenvironment caused by ischemia/reperfusion injury after transplantation, such as over-oxidative stress, nutritional deficiency, and other elements [18]. Therefore, the viability of transplanted MSCs is relatively poor, which offsets the advantages of bone marrow mesenchymal stem cell treatment in controlling ischemic stroke. 

To address this dilemma, we investigated the growth and paracrine function of MSCs in different microenvironments. Results showed that MSCs induced greater growing ability and elevated paracrine function in 10% of ischemic rat brain cultivated MSCs (S-MSCs) (Figure 1 and Figure 2). Our conclusions are in great agreement with a previous study that the trophic factors (BDNF, VEGF, NGF, and HGF) of hMSCs that have been secreted in culture were a response to in vitro therapy with ischemic brain extract [19]. Recent studies have found that acute inflammation promoted angiogenesis and recruitment of progenitor cells, whereas chronic inflammation inhibits the recruitment and survival of local progenitor cells and engrafted MSCs [5]. This study found that the content of the chemokine CRCX4 was significantly increased in S-MSCs, suggesting that the brain tissue homogenate supernatant after ischemic stroke promotes the recruitment of MSCs.

The current experiments showed that the behavior of stroke animals was highly improved after transplantation of MSCs, especially the pretreatment with 10% ischemic rat brain. Our studies showed that S-MSCs have neuroprotective abilities on the MCAO model by decreasing infarct area and restoring neural function. Meanwhile, there are a great many theories about improvement in the behavior of the animal model after MSCs transplantation. The MSCs play a significant role in neural protection and regeneration by inhibiting inflammation [20], decreasing apoptosis [21], releasing neuroprotective or neurotrophic factors, and vascular regeneration [22]. These mechanisms lead to functional recovery by amending neural networks and rebuilding neural circuits. In this paper, S-MSCs decreased positive cells of TUNEL in the ischemic penumbra after stroke, suggesting that the neuroprotective effect of MSCs was not only linked to their function in neurogenesis, angiogenesis, alleviating inflammatory cascade reaction, and formatting of structural reorganization, as previously reported [23,24], but also to their ability in antagonistic apoptotic. Some studies have shown that MSCs transplantation would inhibit neuronal apoptosis by inducing the expression of Bcl-2, a proto-oncogene in a stroke model, which has been proved to regulate the expression of apoptosis [25]. It is worth noting that the expression of Bcl-2 induced by S-MSCs transplantation was higher than that of MSCs transplantation alone. The results were consistent with the finding by TUNEL, which indicated that S-MSCs transplantation has a better effect of concerted inhibition on neuronal apoptosis. In addition, researchers found that the increase in BDNF regulated PI3K/Akt pathway and promoted neurogenesis [26,27]. Xiao’s study found that I/R injury promoted the secretion of exosomes from rat endothelial cells, and those exosomes could significantly improve the I/R injury of SH-SY5Y cells, down-regulate the expression levels of Bax and Caspase-3, and up-regulate the expression of Bcl-2 [28]. It suggests that S-MSCs can improve cerebral ischemic stroke injury by regulating apoptosis-related pathways.

CXCR4 plays a key role in inducing the invasion, extravasation, homing, migration, and survival of stem cells; research has found that induction of CXCR4 expression could promote the increase in BrdU in traumatic brain tissue of rats, that is, promoting the repair of nerve cells in traumatic brain tissue [29]. Our results show that the damage degree has been significantly improved after MSCs treatment, and it was most obvious in S-MSCs. BrdU is inserted into the cell cycle during the S phase of DNA synthesis, so once new cells undergo this step of the synthesis, researchers can use immunological methods to detect the expression of new cells [30]. Therefore, we used BrdU as a marker for newborn nerve cells for detection. From the experimental results, we found that the increase in the S-MSCs group was the most obvious. It suggested that S-MSCs may promote rats’ nerve regeneration after focal cerebral ischemia-reperfusion injury via CXCR4, which possibly is one of the mechanisms of S-MSCs protecting the brain.

Functional and structural destruction of synaptic transmission was the reason for neurological deficits following cerebral ischemia [31]. Recent findings suggest that neuroplasticity and neuronal reorganization play vital roles in functional recovery after stroke [32]. By applying immunohistochemical techniques, this paper focused on GFAP, NeuN, and neuronal NF-L, which can exhibit simultaneous cellular alterations. In this paper, concerning GFAP, which is an astrocyte marker, we observed a reduction in its level in the peri-infarct zone in treated groups with respect to the model group, which is in line with a previous report by Leu et al. [33]. During the reactive process, the astrocytes became hypertrophied and inhibited axonal regeneration to participate in glial scar formation [34]. The NF-L, as a neuronal cytoskeletal element, is vital for maintaining cellular integrity and axonal transport [35]. The immunohistochemical results showed relevant alterations of NF-L after focal cerebral ischemia, which concluded that neurofilaments played a critical role in axonal damage and cytoskeletal degeneration by ischemia [36]. Our results suggested neurogenesis in the ischemic stroke brain was activated, and S-MSCs have a stronger promoting effect. In recent years, studies suggested ischemic stroke via several growth factors promotes neurogenesis, including FGF-2, IGF-1, BDNF, VEGF, and chemokines, including SDF-1, MCP-1 [37]. Additionally, another study proposed that BDNF pretreatment of human embryonic neural stem cells promoted the secretion of VEGF, and together with untreated human embryonic neural stem cells could better improve cell survival and functional recovery after hypoxic-ischemic stroke transplantation [38]. Our study proved that the MSCs, cultured by homogenate supernatant of the ischemic stroke brain, have higher growth factor levels, suggesting that the ischemic brain may promote the differentiation and chemotaxis of MSCs by secreting growth factors, and afford a valued theoretical basis and experimental reference for the treatment of autologous MSCs transplantation in the acute phase of ischemic stroke. Moreover, the secretion of exocrine after brain ischemia injury may promote the repairability of MSCs to brain injury, the biological mechanism of it is worth further study in the future.

## 5. Conclusions

In conclusion, this study provides a relative candidate pretreatment method that can be used for cell transplantation treatment of ischemic cerebral infarction. Our results showed that after MCAO modeling in SD rats, S-MSCs treatment can greatly diminish the neurological deficits and infarct size of the brain. Additionally, the potential mechanism of S-MSC-mediated neuroprotection may involve the up-regulation of BDNF and HGF in the upper ring infarcted cortex to regulate synaptic plasticity and paracrine function. In addition, the results in vivo evidenced that MSC pretreated with S-MSCs significantly enhanced the effect of MSC in reducing apoptosis. On the one hand, pretreatment of ischemia stroke brain may improve the ability of MSC to promote nerve recovery via CXCR4 and BDNF secretion; on the other hand, it inhibited apoptosis by advancing BDNF to regulate Bcl-2 family to improve cell viability and the survival rate. Therefore, S-MSC may be a potentially effective way to treat ischemic stroke. Further research is urgently needed to determine the pathways that lead to the observed therapeutic effect of S-MSC.

## Figures and Tables

**Figure 1 brainsci-12-00631-f001:**
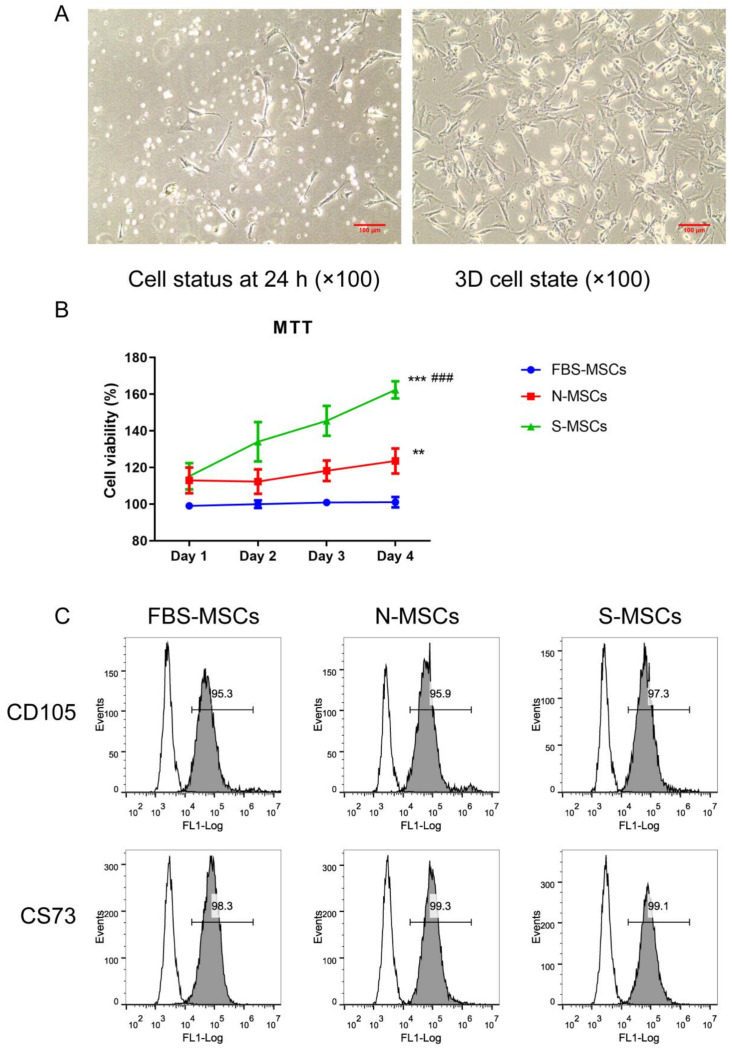
Characteristics of Mmesenchymal stem cells (MSCs): (**A**) Morphological characteristics of MSCs (Scale bar = 100 μm); (**B**) MTT test of MSCs’ viability. Cells were cultured and optical density was serially monitored at 540 nm ($\bar{x}\pm s$, *n* = 3); (**C**) Flow cytometric analysis of MSC surface markers, CD73 and CD105 of MSCs. ** *p* < 0.01, N-MSCs treated group versus FBS-MSCs group. *** *p* < 0.001, S-MSC-treated group versus FBS-MSCs group. ^###^
*p* < 0.001, S-MSC-treated group versus N-MSCs group.

**Figure 2 brainsci-12-00631-f002:**
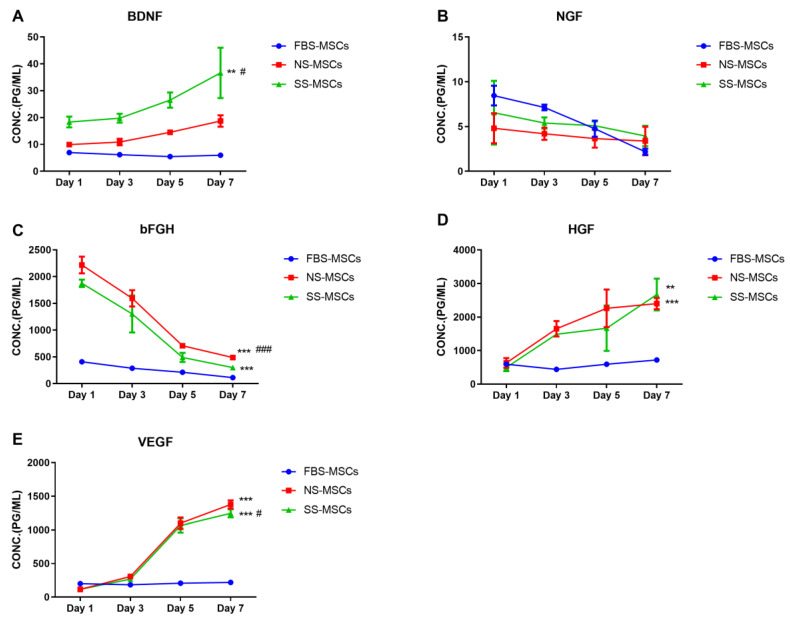
Comparative analysis of paracrine factors secreted by MSCs after pretreatment of rat cerebral infarction tissue ($\bar{x}\pm s$, *n* = 3): (**A**) brain-derived neurotrophic factor (BDNF), (**B**) nerve growth factor (NGF), (**C**) basic fibroblast growth factor (bFGF), (**D**) hepatocyte growth factor (HGF), (**E**) vascular endothelial growth factor (VEGF). ** *p* < 0.01, N-MSCs treated group versus FBS-MSCs group. *** *p* < 0.001, S-MSC group versus FBS-MSCs group. ^#^ *p* < 0.05, ^###^
*p* < 0.001, S-MSC-treated group versus N-MSCs group.

**Figure 3 brainsci-12-00631-f003:**
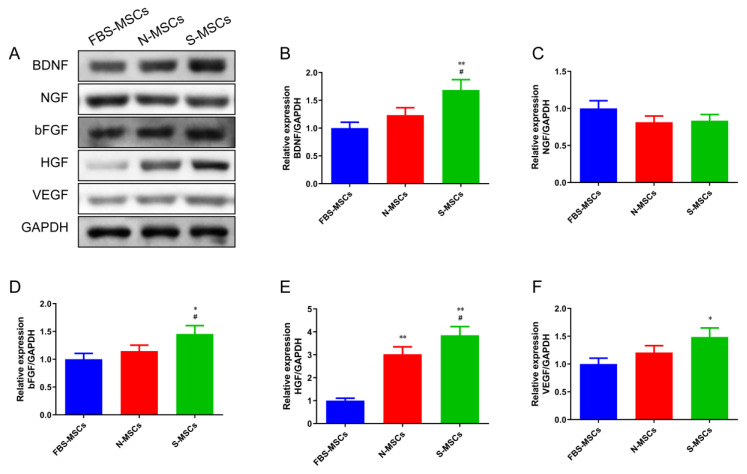
Comparative analysis of paracrine factors secreted by MSCs after pretreatment of rat cerebral infarction tissue: (**A**) Western blot analysis of the expression of brain-derived neurotrophic factor (BDNF) (**B**), nerve growth factor (NGF) (**C**), basic fibroblast growth factor (bFGF) (**D**), (hepatocyte growth factor) HGF (**E**), and vascular endothelial growth factor (VEGF) (**F**) in MSCs cultured with FBS, normal ischemic brain tissue, or ischemic brain tissue ($\bar{x}\pm s$, *n* = 3). * *p* < 0.05, ** *p* < 0.01, compared with FBS-MSCs group; ^#^
*p* < 0.05, compared with N-MSCs group.

**Figure 4 brainsci-12-00631-f004:**
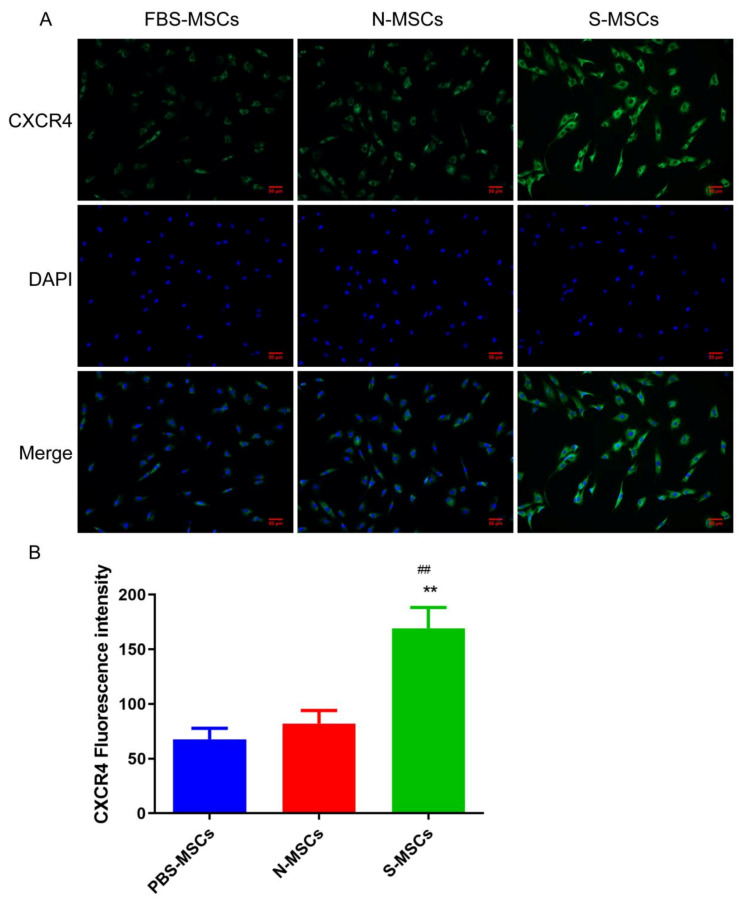
(**A**) Immunofluorescence detection of the expression of CXCR4 on the cell surface (×200), (Scale bar = 50 μm). (**B**) Quantitative results of cell surface chemokine CXCR4 expression ($\bar{x}\pm s$, *n* = 6). Compared with FBS-MSCs group, ** *p* < 0.01. Compared with N-MSCs group, ^##^
*p* < 0.01.

**Figure 5 brainsci-12-00631-f005:**
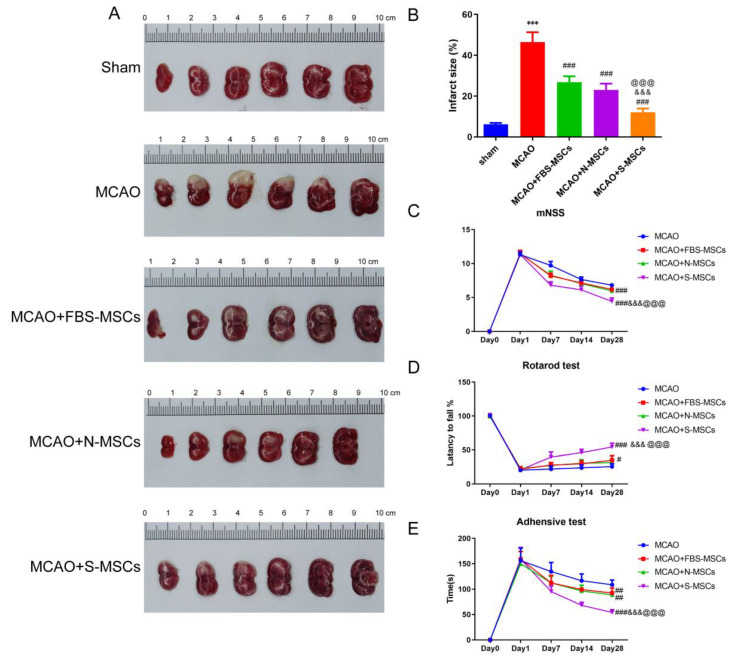
S-MSCs reduced the cerebral infarction volume and alleviated neurologic deficits in ischemia/reperfusion (I/R) rats: (**A**) Cerebral infarct area in MCAO rats (TTC staining). (**B**) Quantitative analysis of the cerebral infarct volume. Results of behavioral tests. (**C**) mNSS scores. (**D**) The rotarod test. (**E**) The adhesive-removal test. Compared with the sham group, *** *p* < 0.001; compared with the MCAO group, ^#^
*p* < 0.05, ^##^
*p* < 0.01 and ^###^
*p* < 0.001; compared with the MCAO + FBS-MSCs group, ^&&&^
*p* < 0.01; compared with MCAO + N-MSCs group, ^@@@^
*p* < 0.001. ($\bar{x}\pm s$, *n* = 6).

**Figure 6 brainsci-12-00631-f006:**
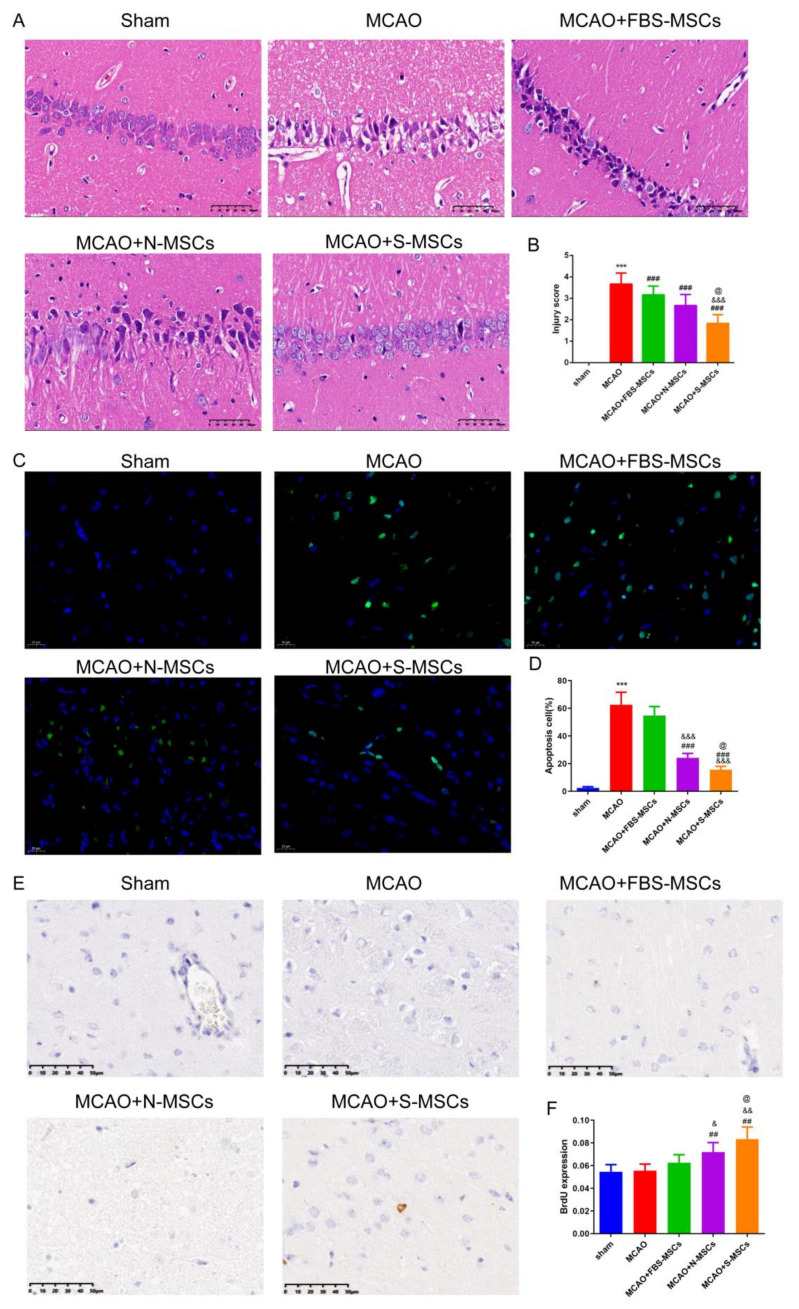
(**A**) Brain tissue recovery in MCAO rats (HE staining, ×400). (**B**) Quantitative analysis of the injury score using ImageJ software. (**C**) TUNEL double staining was performed to evaluate neuronal apoptosis in slices from right ischemic cortices (Scale bar = 100 μm). (**D**) Quantitative analysis of the AI using ImageJ software. (**E**) Detection of BrdU expression in brain tissue of rats with ischemia-reperfusion cerebral infarction (×400). (**F**) Quantitative analysis of the BrdU using ImageJ software. Compared with the sham group, *** *p* < 0.001; compared with the MCAO group, ^##^
*p* < 0.01, ^###^
*p* < 0.001; compared with the MCAO + FBS-MSCs group, ^&^
*p* < 0.05, ^&&^
*p* < 0.01, ^&&&^
*p <* 0.001; compared with MCAO + N-MSCs group, ^@^
*p* < 0.01. ($\bar{x}\pm s$, *n* = 6).

**Figure 7 brainsci-12-00631-f007:**
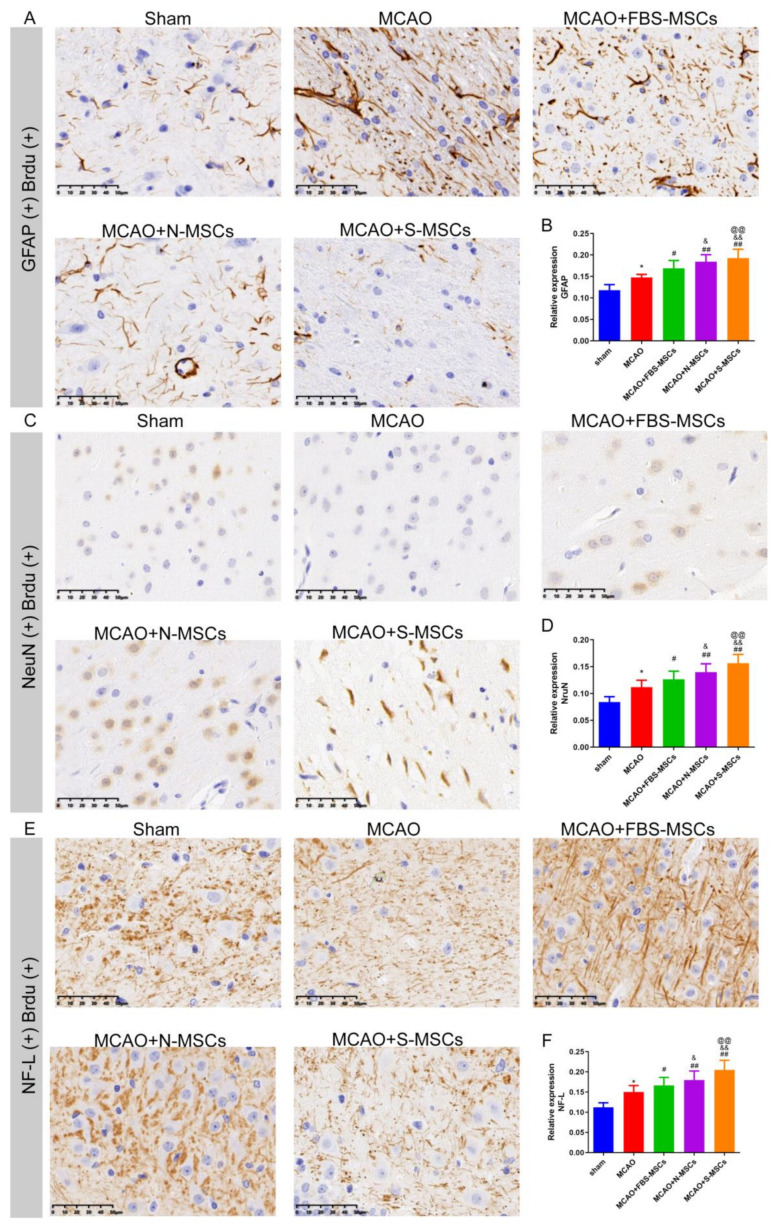
Immunohistochemical detection of GFAP/NeuN/NF-L expression level in brain tissue of rats (×400). ($\bar{x}\pm s$, *n* = 6) (**A**,**B**) Immunohistochemical co-localization of GFAP and Brdu; (**C**,**D**) Immunohistochemical co-localization of NeuN and Brdu; (**E**,**F**) Immunohistochemical co-localization of NF-L and Brdu. Compared with the sham group, * *p* < 0.05; compared with the MCAO group, ^#^
*p* < 0.05, ^##^
*p* < 0.01; compared with the MCAO + FBS-MSCs group, ^&^
*p* < 0.05, ^&&^
*p* < 0.01; compared with the MCAO + N-MSCs group, ^@@^
*p* < 0.01.

**Figure 8 brainsci-12-00631-f008:**
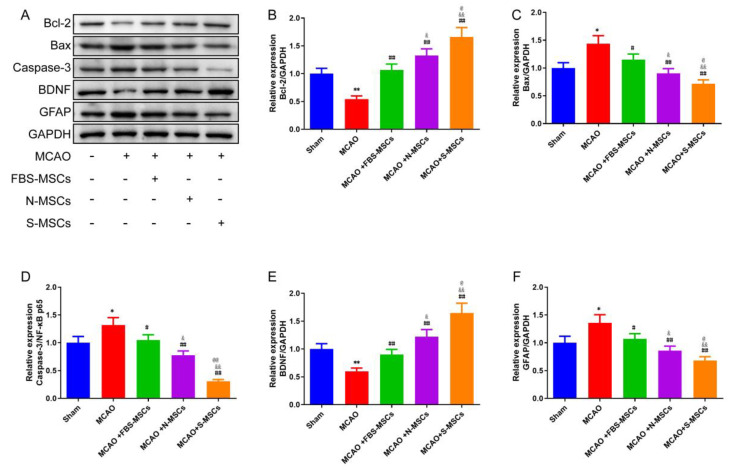
(**A**) Western blot analysis of the rat hippocampus neurons (**B**) Bcl-2, (**C**) Bax, (**D**) Caspase-3, (**E**) BDNF, and (**F**) GFAP proteins expression ($\bar{x}\pm s$, *n* = 3). Compared with the sham group, * *p* < 0.05, ** *p* < 0.01; compared with the MCAO group, ^#^
*p* < 0.05, ^##^
*p* < 0.01; compared with the MCAO +FBS-MSCs group, ^&^
*p* < 0.05, ^&&^
*p* < 0.01; compared with the MCAO +N-MSCs group, ^@^
*p* < 0.05, ^@@^
*p* < 0.01.

## Data Availability

Not applicable.

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
