# Peer review of "Preconditioned MSCs Alleviate Cerebral Ischemia-Reperfusion Injury in Rats by Improving the Neurological Function and the Inhibition of Apoptosis"

_brainsci, 2022, doi:10.3390/brainsci12050631_

Round 1
Reviewer 1 Report
Unfortunately, although subject and overall idea of the research is interesting manuscript is impossible to read. English language is very poor- sentence construction, grammar (tenses, passive & active, interpunction).
It is hard to follow Material and methods. Quality of the Figures is very low...
After correction of all parts of the manuscript (with emphasis on English language, quality of images,...) the manuscript could be revised.
Author Response
Dear Reviewer 1,
Thank you very much for your hard work in reviewing the manuscript “Preconditioned MSCs alleviate cerebral ischemia-reperfusion injury in rats by improving the neurological function and the inhibition of apoptosis” (manuscript ID: brainsci-1654383). We have revised the manuscript, according to the reviewers’ comments. The change of manuscript has been marked up using the simple mark of the “Track Changes” function.
Point 1: Unfortunately, although subject and overall idea of the research is interesting manuscript is impossible to read. English language is very poor-sentence construction, grammar (tenses, passive & active, interpunction).
It is hard to follow Material and methods. Quality of the Figures is very low...
After correction of all parts of the manuscript (with emphasis on English language, quality of images,...) the manuscript could be revised.
Response 1: The co-anthers and I would like to thank you for the time and effort spent on reviewing our manuscript. Also, we deeply appreciate you for your recognition of our subject and the overall idea of the research. We have carefully reconsidered and revised the language, which has also been checked by native English speakers. And we replaced the pictures with higher quality. The change of manuscript has been marked up using the “Track Changes” function and will not influence the content and framework of the paper. Hope that the correction will meet with approval.
All the above changes will not influence the content and framework of the paper. We appreciate for Editors and Reviewers’ warm work earnestly and hope that the correction will meet with approval. Once again, thank you very much for your comments and suggestions.
Sincerely
Jin Zheng

Reviewer 2 Report
- Was whole brain extract (in N-MSC, S-MSC) used to stimulate MSC?
- How many MSCs were transplanted into mice in each group?
- It seems that the growth factor level of N-MSC is also relatively high. Please discuss your suggestions on it.
- It would be recommended that the number of mice in each group in each figure is in the legend.
- Did you observe any significant differences in survival rates among experimental groups?
Author Response
Dear Reviewer 2,
Thank you very much for your careful review of the manuscript “Preconditioned MSCs alleviate cerebral ischemia-reperfusion injury in rats by improving the neurological function and the inhibition of apoptosis” (manuscript ID: brainsci-1654383). We have revised the manuscript, according to the reviewers’ comments. The change of manuscript has been marked up using the simple mark of the “Track Changes” function. We would be very grateful if the manuscript could be published in brain science. The following part is the point-by-point responses to the comments.
Point 1: Was whole brain extract (in N-MSC, S-MSC) used to stimulate MSC?
Response 1: Thank you for your hard work. The whole-brain extract was used to stimulate MSC. We have modified its description in Section 2.4 of "Materials and methods", hoping to meet the requirements. (Lines 147-157)
Point 2: How many MSCs were transplanted into mice in each group?
Response 2: We gratefully appreciate for your valuable suggestion. The 2´106 MSCs were transplanted into mice in each group. (Lines 187)
Point 3: It seems that the growth factor level of N-MSC is also relatively high. Please discuss your suggestions on it.
Response 3: Thank you. We agree with the reviewer that the growth factor level of N-MSC is also relatively high in Figure 2 by ELISA. Some researchers found that N-MSCs have effects of neuroprotective, However, their poor survival after transplantation limits their therapeutic effects and applications. In this paper, we analyzed paracrine factors secreted by MSCs after pretreatment of rat cerebral infarction tissue by Western blot (Figure 3), found that compared with the FBS-MSCs group, the protein levels of BDNF, NGF, bFGF and VEGF in the N-MSCs group was no statistical difference, the protein level of HGF was markedly increased. The levels of BDNF, bFGF, HGF and VEGF proteins in the S-MSCs group were significantly growing (Lines 352-360). In particular, we found that there was a large difference in the content of BDNF between the N-MSCs group and the S-MSCs group, which may be related to the S-MCSs treatment in MCAO rats to enhance the ability of MSCs to promote the transplantation success and nerve recovery.
Point 4: It would be recommended that the number of mice in each group in each figure is in the legend.
Response 4: It is really a great suggestion as the Reviewer pointed out that we should mark the number of rats in each group in the legend. According to this suggestion, we have made modifications. (Lines 335, 364, 375, 390, 420, 459, 475, and 482)
Point 5: Did you observe any significant differences in survival rates among experimental groups?
Response 5: Thank you. Before the experiment, we eliminated the rats that died due to the failure of modeling. The survival rate of MCAO rats is about 90%. And finally, after successfully modeling, 6 rats in each group were taken for subsequent experiments and materials.
All the above changes will not influence the content and framework of the paper. We appreciate for Editors and Reviewers’ warm work earnestly and hope that the correction will meet with approval. Once again, thank you very much for your comments and suggestions.
Sincerely
Jin Zheng

Reviewer 3 Report
In this study, the authors examined whether culture conditions of mesenchymal stem cells (MSCs) affect their survival and therapeutic effects on cerebral ischemic injury. The results showed that cultivation of MSCs in the presence of ischemic brain tissue could enhance therapeutic effects via upregulation of some proteins related to neuroprotection after ischemic stroke. Finally, the authors concluded that pretreatment (precondition) of MSCs with ischemic brain tissue after a stroke might become a strategy for treating ischemic stroke. However, some points are needed to be addressed before publication in Brain Sciences.
1) Cultivation of MSCs under 10% ischemic brain tissue was found to increase therapeutic benefits by MSC treatment in MCAO rats. What is the underlying mechanism to improve the functionalities of MSCs by co-culture with ischemic brain tissue or normal brain tissue, compared with FBS? What is involved? Were any nutritional factors (such as extracellular vesicles) secreted from brain tissue, and are there any possibilities of involvement of those factors for enhancement of MSC function? The authors should indicate not only the therapeutic effects but also the underlying mechanisms of effects of preconditions with some data.
2) The results of TUNEL staining were shown in Figure 6. However, there were no explanation regarding TUNEL staining in Material and methods. The authors should clearly mention. Also, what is apoptosis index (AI)?
3) The resolution of Figures is too poor throughout the manuscript, so the authors should improve.
4) Several abbreviations were used without writing full-spelling. Also, there are a lot of English errors. Please check.
Author Response
Dear Reviewer 3,
Thank you very much for your careful review of the manuscript “Preconditioned MSCs alleviate cerebral ischemia-reperfusion injury in rats by improving the neurological function and the inhibition of apoptosis” (manuscript ID: brainsci-1654383). We have revised the manuscript, according to the reviewers’ comments. The change of manuscript has been marked up using the simple mark of the “Track Changes” function. We would be very grateful if the manuscript could be published in brain science. The following part is the point-by-point responses to the comments.
Point 1: Cultivation of MSCs under 10% ischemic brain tissue was found to increase therapeutic benefits by MSC treatment in MCAO rats. What is the underlying mechanism to improve the functionalities of MSCs by co-culture with ischemic brain tissue or normal brain tissue, compared with FBS? What is involved? Were any nutritional factors (such as extracellular vesicles) secreted from brain tissue, and are there any possibilities of involvement of those factors for enhancement of MSC function? The authors should indicate not only the therapeutic effects but also the underlying mechanisms of effects of preconditions with some data.
Response 1: We deeply appreciate for this good and professional suggestion, the underlying therapeutic mechanism should also be discussed, and we have indicated this content in our revised manuscript. In the discussion part, we added a discussion on the action mechanism. The specific contents are as follows “In recent years, studies proposed that BDNF pretreatment of human embryonic neural stem cells promoted the secretion of VEGF, and together with untreated human embryonic neural stem cells could better improve cell survival and functional recovery after hypoxic-ischemic stroke transplantation[1]. Meng’s team found that promoting the SDF-1/CXCR4 axis and BDNF expression could promote the recovery of neural function after spinal cord injury in rats[2]. Furthermore, studies have proved that small extracellular vesicles obtained from hypoxic mesenchymal stromal cells can better promote angiogenesis and nerve recovery[3]. It suggested that pretreatment of MSCs by ischemic stroke brain of rats may have the potential to activate the secretion of growth factors, especially neural-related, and also promote the survival and function of MSCs, thus this treatment in MCAO rats may enhance the ability of MSCs to promote the transplantation success and nerve recovery.” (Lines 612-627) “In addition, researchers found that the increase of BDNF regulated PI3K/Akt pathway and promoted neurogenesis[4, 5]. Xiao's study found that I/R injury promoted the secretion of exosomes from rat endothelial cells, and those exosomes could significantly improve the I/R injury of SH-SY5Y cells, down-regulated the expression levels of Bax and Caspase-3, and up-regulate the expression of BCL-2[6]. Whether the secretion of exocrine after brain ischemia injury promotes the repair ability of MSCs to brain injury is worth further study in the future.” (Lines 648-657)
Point 2: The results of TUNEL staining were shown in Figure 6. However, there were no explanation regarding TUNEL staining in Material and methods. The authors should clearly mention. Also, what is apoptosis index (AI)?
Response 2: We are very sorry that we have omitted the method description of TUNEL. According to the comments of reviewers, we have supplemented this part in the method part. The specific description is as follows “The cells were seeded on the cover glass with a cell density of 60-70%, and the cells were treated according to grouping. Then discarded the culture medium and washed the cells. Add 4% paraformaldehyde to fix the cells for 30 minutes and soak them in PBS for 5 minutes. After permeating the cell membrane with 0.3% Triton X-100 (Hyclone, USA), the cells were stained with the TUNEL kit (Beyotime, China). Between each step, soaked in PBS 3 times for 5 minutes each time. Finally, the slides were sealed and observed under an inverted fluorescence microscope. Apoptosis rate %: the ratio of TUNEL positive cells to total cells.” (Lines 272-282)
Point 3: The resolution of Figures is too poor throughout the manuscript, so the authors should improve.
Response 3: Thank you for your precious comments and advice. We have promoted the quality of the figures in the manuscript to a higher pixel.
Point 4: Several abbreviations were used without writing full-spelling. Also, there are a lot of English errors. Please check.
Response 4: We apologize for the language problems in the original manuscript. We have carefully reconsidered and revised the language, which has been checked by native English speakers. The change of manuscript has been marked up using the “Track Changes” function.
All the above changes will not influence the content and framework of the paper. We appreciate for Editors and Reviewers’ warm work earnestly and hope that the correction will meet with approval. Once again, thank you very much for your comments and suggestions.
Sincerely
Jin Zheng
References:
- Rosenblum, S., et al., BDNF Pretreatment of Human Embryonic-Derived Neural Stem Cells Improves Cell Survival and Functional Recovery After Transplantation in Hypoxic-Ischemic Stroke. Cell Transplant, 2015. 24(12): p. 2449-61.
- Meng, X.L., et al., Hyperbaric oxygen improves functional recovery of rats after spinal cord injury via activating stromal cell-derived factor-1/CXC chemokine receptor 4 axis and promoting brain-derived neurothrophic factor expression. Chin Med J (Engl), 2019. 132(6): p. 699-706.
- Dumbrava, D.A., et al., Mesenchymal stromal cell-derived small extracellular vesicles promote neurological recovery and brain remodeling after distal middle cerebral artery occlusion in aged rats. Geroscience, 2022. 44(1): p. 293-310.
- Zhu, T., et al., Notoginsenoside R1 Improves Cerebral Ischemia/Reperfusion Injury by Promoting Neurogenesis via the BDNF/Akt/CREB Pathway. Front Pharmacol, 2021. 12: p. 615998.
- Li, Y., et al., Protection against acute cerebral ischemia/reperfusion injury by Leonuri Herba Total Alkali via modulation of BDNF-TrKB-PI3K/Akt signaling pathway in rats. Biomed Pharmacother, 2021. 133: p. 111021.
- Xiao, B., et al., Endothelial cell-derived exosomes protect SH-SY5Y nerve cells against ischemia/reperfusion injury. Int J Mol Med, 2017. 40(4): p. 1201-1209.

Round 2
Reviewer 1 Report
Authors aimed to investigate whether ischemic microenvironment could improve mesenchymal stem cells (MSCs) survival and secretion of growth factors. Further goal was to establish whether those pretreated MSCs could better alleviate cerebral ischemic injury in rats. For that purpose, Authors performed in vitro experiment with MSCs cultured in FBS, 10% normal brain tissue and 10% ischemic brain tissue and then in vivo experiment where they transfer those MSCs in MCAO rats. Authors observed survival rate of cultured MSCs and level of secreted growth factors and other molecules, and neurological score, apoptotic molecules, neurogenesis etc. in rat MCAO model.
Experimental design is very interesting, experiments were designed appropriately. Unfortunately there is one major obstacle for publishing this manuscript.
Although Authors emphasized that manuscript is revised by English speaker current version is still unacceptable. In addition to grammatical irregularities and incorrect sentence constructions there are major mistakes like 'coronary brain slices' instead of 'coronal brain slices', misuse of terms 'treatment', 'treated' ...etc. Please improve the whole manuscript, in this form it is not acceptable.
Because of that it is hard to read and see the scientific part of the manuscript.
Introduction
Please provide at the end short rationale for using parameters presented in the manuscript.
M&M
please divide clearly in vitro and in vivo part of study...Authors first described MCAO and then cell culture. Some parts of M&M are copied from protocol- they should be re-written. one example (of many)
'After removing the arteriole clip, the ophthalmic forceps Line plug was slowly advanced. When the front of the line-of-sight plug enters the ICA through the CCA bifurcation, adjust the curve of the plug line to the lower right direction.' it should be 'the curve of the plug was adjusted...'
Authors write about paraffin-embedded tissue but at the beginning they wrote coronary frozen sections. What regions did they analyse?
Did Authors performed TUNEL assay on tissue slices? In the manuscript 'The cells were seeded on the cover glass with a cell density of 60-70%, and the cells were treated according to grouping. '
Discussion
Too little actual discussion.
Authors mostly described results again..and wrote some well known facts (such as role of BrdU...etc). What about real discussion about results and about already published data from other studies..
Author Response
Response to Reviewer 1 Comments
Dear Reviewers,
Thank you for your letter and for the comments on our manuscript entitled “Preconditioned MSCs alleviate cerebral ischemia-reperfusion injury in rats by improving the neurological function and the inhibition of apoptosis” (manuscript number: brainsci-1654383). We are sorry that our lack of language has caused you trouble reading. We have revised the Introduction, Materials and methods, and Discussion of the manuscript according to these comments. All changes to the manuscript have been marked up using the simple mark of the “Track Changes” function. Hope this manuscript can meet your expectations. Of course, because the revision time is short, we will further revise the language of the manuscript with the help of a professional English editor after this submission. Thank you again for your sincere advice, which greatly helps to improve the readability of our manuscript. The following part is the point-by-point responses to the comments.
Point 1: Authors aimed to investigate whether ischemic microenvironment could improve mesenchymal stem cells (MSCs) survival and secretion of growth factors. Further goal was to establish whether those pretreated MSCs could better alleviate cerebral ischemic injury in rats. For that purpose, Authors performed in vitro experiment with MSCs cultured in FBS, 10% normal brain tissue and 10% ischemic brain tissue and then in vivo experiment where they transfer those MSCs in MCAO rats. Authors observed survival rate of cultured MSCs and level of secreted growth factors and other molecules, and neurological score, apoptotic molecules, neurogenesis etc. in rat MCAO model.
Experimental design is very interesting, experiments were designed appropriately. Unfortunately there is one major obstacle for publishing this manuscript.
Although Authors emphasized that manuscript is revised by English speaker current version is still unacceptable. In addition to grammatical irregularities and incorrect sentence constructions there are major mistakes like 'coronary brain slices' instead of 'coronal brain slices', misuse of terms 'treatment', 'treated' ...etc. Please improve the whole manuscript, in this form it is not acceptable.
Because of that it is hard to read and see the scientific part of the manuscript.
Response: Thank you for your careful review and constructive comments. We have checked and revised the language of the manuscript, and all changes have been made using the “Track Changes” function and presented in the manuscript with simple markup. In addition, due to the short revision time of this submission, it is not enough for us to apply for the language polishing at a professional institution. So, we accepted the Managing Editor’s suggestion that we could polish the language of the manuscript with the help of the English Editor at the proofreading stage.
Point 2: Introduction.
Please provide at the end short rationale for using parameters presented in the manuscript.
Response: Thank you for your comments. According to this suggestion, we have made the following changes to the last paragraph of the Introduction, “Our research team used rat cerebral infarction tissue to pretreat rat MSCs, evaluating the reproductive capacity, survival rate, the levels of nutritional factors and CXCR4 of the MSCs in vitro. Afterward, the behavioral tests of ischemic stroke rats and measurements of infarct area by TTC were used to evaluate the neuronal injuries of rats and the protective effect of pretreated MSCs. And the degree of pathological injury and apoptosis in the cerebral infarction area of stroke rats was evaluated by HE and TUNEL staining. The levels of proliferation, differentiation, and injury in neurons were observed by immunohistochemistry. In addition, we detected the apoptosis-related proteins and the nutritional factors by WB to explore the potential mechanism of MSCs to improve ischemic stroke. This paper hopes to afford a valued theoretical basis and experimental reference for the treatment of autologous MSCs transplantation in the acute phase of ischemic stroke.” (Lines 86-97)
Point 3: M&M
please divide clearly in vitro and in vivo part of study...Authors first described MCAO and then cell culture. Some parts of M&M are copied from protocol- they should be re-written. one example (of many)
'After removing the arteriole clip, the ophthalmic forceps Line plug was slowly advanced. When the front of the line-of-sight plug enters the ICA through the CCA bifurcation, adjust the curve of the plug line to the lower right direction.' it should be 'the curve of the plug was adjusted...'
Response: These comments were helpful to us. We have revised the method description on “Materials and methods” and all changes to the manuscript have been marked up using the simple mark of the “Track Changes” function. In addition, we have moved the description of the cell experiments backward. 2.1-2.9 are the part of animal experiments (Lines 102-257); 2.10-2.15 are the part of cell experiments (Lines 259-320); 2.16 Western blot was used on animals and cells (Lines 321-343).
Point 4: Authors write about paraffin-embedded tissue but at the beginning they wrote coronary frozen sections. What regions did they analyse?
Response: We are very sorry that our carelessness has troubled you. We did not use any frozen sections. All sections were paraffin-embedded. And we have corrected this mistake in the manuscript. (Lines 221)
Point 5: Did Authors performed TUNEL assay on tissue slices? In the manuscript 'The cells were seeded on the cover glass with a cell density of 60-70%, and the cells were treated according to grouping. '
Response: Sorry for this error, we only performed TUNEL staining on rat tissue slices. And we have revised the TUNEL staining (Lines 250-258).
Point 6: Discussion
Too little actual discussion.
Authors mostly described results again..and wrote some well known facts (such as role of BrdU...etc). What about real discussion about results and about already published data from other studies.
Response: Thank you for your comments. We have checked and revised the Discussion, and all changes have been made using the “Track Changes” function and presented in the manuscript with simple markup. (Lines 560-578, 588-611, 616-619, 624-628, 639, 648-668)
We appreciate for Reviewer’s warm work earnestly and hope that the correction will meet with approval. Once again, thank you very much for your comments.
Sincerely
Jin Zheng
Reviewer 3 Report
The authors significantly improved the manuscript.
Author Response
Response to Reviewer 3 Comments
Dear Reviewer,
Thank you for your letter and for the comments on our manuscript entitled “Preconditioned MSCs alleviate cerebral ischemia-reperfusion injury in rats by improving the neurological function and the inhibition of apoptosis” (manuscript number: brainsci-1654383). We hope that the manuscript could be published in brain science.
Point 1: Comments and Suggestions for Authors
The authors significantly improved the manuscript.
Response: Thank you for your recognition of us, we hope this revision can meet your requirements.
We appreciate for Reviewer’s warm work earnestly and hope that the correction will meet with approval. Once again, thank you very much for your comments.
Sincerely
Jin Zheng